# Orofacial Migraine and Neurovascular Orofacial Pain: Response to Treatment—A Pilot Study

**DOI:** 10.3390/biomedicines13030714

**Published:** 2025-03-14

**Authors:** Rafael Benoliel, Yair Sharav, Shimrit Heiliczer, Yaron Haviv

**Affiliations:** 1Department of Diagnostic Sciences, Rutgers School of Dental Medicine, Newark, NJ 07103, USA; benolira@sdm.rutgers.edu; 2Department of Oral Medicine, Sedation and Imaging, Hadassah Medical Center, Faculty of Dental Medicine, Hebrew University of Jerusalem, Jerusalem 91120, Israel; sharavy@mail.huji.ac.il; 3Oral Medicine Unit, Oral and Maxillofacial Surgery Department, Tel Aviv Sourasky Medical Center, Tel Aviv 6423906, Israel; hsaroni@yahoo.com

**Keywords:** orofacial migraine, neurovascular orofacial pain, International Classification of Orofacial Pain, response to pharmacotherapy

## Abstract

**Introduction:** The International Classification of Orofacial Pain (ICOP) recognizes orofacial migraine (OFM) and neurovascular orofacial pain (NVOP) as migraine-related entities affecting the facial and oral regions. The diagnostic features of OFM and NVOP indicate that there are many similarities between the two. However, we recently demonstrated that NVOP and OFM are two distinct diagnostic entities, confirming the ICOP classification. It was the aim of the present study to examine whether OFM and NVOP differ in response to pharmacotherapy. **Materials and Methods**: The cohort was made up of 40 patients attending a tertiary orofacial pain clinic. When implementing ICOP criteria, an OFM diagnosis was made in 23 and an NVOP diagnosis in 17. **Results**: No statistically significant differences between NVOP versus OFM were observed in the global response to standard abortive therapy such as triptans, or NSAIDs. Similarly, no statistically significant differences were found following prophylactic therapy that included beta-blockers, anti-epileptic drugs, and tricyclic antidepressants. Up to 80% of patients responded favorably with ≥50% pain reduction. **Conclusions**: NVOP and OFM differ in diagnostic characteristics, demonstrating unique features, and were confirmed as two diagnostic entities. However, NVOP and OFM did not differ in their response to abortive or prophylactic treatments. Study limitations include the lack of starting data precluding a more precise pharmacological analysis. The small sample size limits any far reaching conclusions. This is particularly true regarding individual drug efficacy. We were unable to analyze drug and dose responses separately due to data constraints.

## 1. Introduction

Primary headaches presenting in the facial region have been recognized and recently classified in the International Classification of Orofacial Pain, 1st edition (ICOP) [1]. Section 5 in ICOP defines “orofacial pains resembling presentations of primary headaches”, which covers four main sub-sections; orofacial migraine, tension-type orofacial pain, trigeminal autonomic orofacial pain, and neurovascular orofacial pain. Of interest to the present study are the responses to pharmacotherapy of patients with orofacial migraine (OFM) and neurovascular orofacial pain (NVOP). According to ICOP, [1] OFM is subdivided into episodic and chronic types. Both occur exclusively in the orofacial region, without head pain, with the characteristics and associated features of migraine as described in ICHD-3 [2]. The episodic type is characterized by recurrent attacks lasting 4–72 h. Typical characteristics of OFM are unilateral location, pulsating quality, moderate or severe intensity, aggravation by routine physical activity, and association with nausea and/or vomiting, photophobia, and phonophobia. The chronic type is characterized by facial and/or oral pain occurring on 15 or more days per month for more than 3 months, which has the features of migraine on at least 8 days per month. The definition of neurovascular orofacial pain (NVOP) by the ICOP [1] includes attacks of variable duration of moderate or severe intraoral pain, without head pain, often accompanied by toothache-like symptoms, with mild cranial autonomic and/or migraine symptoms. NVOP is subdivided by ICOP into short-lasting and long-lasting neurovascular orofacial pain [1]. This diagnostic entity was first described by us in 1997 [3] and was followed by additional reports and studies [4,5]. NVOP typically presents with accompanying tooth-like pain aggravated by cold food or beverages. This is very similar to symptoms of dental pathology, such as a carious lesion, except that these teeth are intact on clinical and radiologic examination. Many NVOP cases are characterized by a high frequency, daily pattern of spontaneous pain or evoked by cold food ingestion, pain is strong (7–8 on VAS), pulsating and episodic. Pain may last minutes to hours and up to 3 days [6].

The individual features of NVOP and OFM were recently reviewed with a focus on whether these two diagnostic entities should be merged or kept as two separate diagnoses [7]. It was concluded that the differences between NVOP and OFM are subtle, and their separation into two entities warrants further investigation. NVOP and OFM patients do not differ in age, sex, pain intensity, and other pain characteristics [7]. Nevertheless, there are significantly more cranial autonomic symptoms in OFM (36.4%) than in NVOP (10.3%) patients. OFM patients have more migraine symptoms such as nausea and photophobia (68.6% vs. 41%). Additionally, OFM patients presented with significantly more pain-related awakening (52.9%) than NVOP patients (26.3%). OFM with an associated headache occurred in 66.7% of cases, compared to 30.8% in NVOP cases. Most NVOP patients report toothache-like pain (85%), rarely detected in OFM (11.4%) [7]. It was concluded that the diagnostic features of OFM and NVOP, while demonstrating many similarities, have sufficient unique features to allow separating them into two distinct diagnostic entities, in accordance with the ICOP classification [7]. However, their response to therapy seems to be similar [8,9,10,11,12,13,14,15] and suggests a need to re-examine the relationship between the two entities.

In the present study, we aim to examine the responses to abortive and prophylactic pharmacotherapy between patients with orofacial migraine (OFM) and neurovascular orofacial pain (NVOP).

## 2. Materials and Methods

### 2.1. Study Design and Population

This retrospective study was conducted at the Orofacial Pain Clinic, Faculty of Dentistry, Hadassah-The Hebrew University, Jerusalem, Israel. The study analyzed a cohort of 40 patients diagnosed with either Orofacial Migraine (OFM, n = 23) or Neurovascular Orofacial Pain (NVOP, n = 17) based on the International Classification of Orofacial Pain (ICOP) criteria [1]. Patients were recruited between 2020 and 2023. The Hadassah Medical Center Helsinki committee (HMO-0473-14) approved this retrospective study. No changes were made to patient treatment or management, and therefore, informed consent from each patient was not required.

### 2.2. Data Collection

Standardized intake forms were used to collect patient demographics, medical history, and detailed pain characteristics. Pain location and quality were assessed using structured diagrams and descriptive terms (e.g., pulsating, stabbing, burning). Additionally, data on migraine-associated symptoms, including nausea, photophobia, phonophobia, and cranial autonomic symptoms (CAS), were documented. A thorough history of possible triggers was taken, including dietary, environmental, and physiological factors. Pain intensity was recorded using an 11-point Visual Analog Scale (VAS).

### 2.3. Clinical Examination

All patients underwent a comprehensive intraoral and extra oral examination to rule out dental, periodontal, and mucosal pathologies. Imaging studies, including panoramic radiographs and periapical imaging, were conducted when necessary to exclude dental pathology. Neurological and musculoskeletal assessments were also performed to evaluate trigeminal nerve function and myofascial involvement.

### 2.4. Inclusion and Exclusion Criteria

Patients included in the study met the ICOP diagnostic criteria [1] for OFM or NVOP. Patients with concurrent head pain were included if their orofacial symptoms met the diagnostic requirements for OFM or NVOP. Exclusion criteria included the presence of significant dental pathology, temporomandibular disorders (TMD), systemic neurological diseases, or a history of medication overuse headache. Patients with incomplete medical records or those undergoing concurrent treatments for other pain disorders were also excluded.

### 2.5. Treatment Protocols

Treatment regimens included both abortive and prophylactic therapies based on established migraine management guidelines. Abortive therapy consisted of triptans (e.g., sumatriptan, rizatriptan) and NSAIDs (e.g., naproxen, ibuprofen). Prophylactic therapies included beta-blockers (e.g., propranolol), anticonvulsants (e.g., topiramate), tricyclic antidepressants (e.g., amitriptyline), and CGRP inhibitors in cases of refractory pain. Medication selection was tailored based on patient history, comorbidities, and medication tolerance. Patients were instructed on non-pharmacological strategies, including dietary modifications, sleep hygiene, and stress management techniques.

### 2.6. Treatment Response Assessment

Patients were followed over a minimum of three months. Responses to treatment were assessed based on a reduction in attack frequency, severity, and duration. Treatment response was classified into two groups: moderate (<50% improvement) or significant (≥50% improvement) based on patient-reported pain diaries and clinical evaluations. Additional factors influencing treatment response, including medication adherence, side effects, and the need for combination therapy, were documented.

### 2.7. Statistical Analysis

Data collected were tabulated and analyzed in SPSS (SPSS Statistics V29, IBM, New York, NY, USA), with a two-tailed α for significance set at 0.05. Missing data for individual variables were coded within SPSS Statistics V29 that adjusted the sample size for analysis accordingly. The null hypothesis was that there would be no differences in clinical response to standard treatment between OFM and NVOP. Associations between diagnosis and other nominal variables were analyzed with the Pearson Chi squared (χ2) test, and a two-sided Student *t*-test for continuous variables. Relevant results are expressed as the mean and standard deviation

## 3. Results

The cohort was made up of 40 patients, with an orofacial migraine (OFM) diagnosis in 23 and a neurovascular orofacial pain (NVOP) diagnosis in 17. OFM patients were on average 37.8 ± 17.2 years and NVOP 40.1 ± 13.0 years old. The OFM group was made up of 19 females (37.7 ± 18.0 years) and 4 males (38.3 ± 16.0 years) and the NVOP group of 14 females (41.3 ± 121.3 years) and 3 males (34.3 ± 17.6 years). The pain intensity rating on an 11-point visual analog scale at presentation was 7.73 ± 1.8 in NVOP and 8.6 ± 1.3 in OFM (t = −1.6, df = 1, *p* = 0.1), see Table 1.

There were no statistically significant differences between NVOP and OFM in laterality, a history of migraine, a throbbing quality to pain, waking from sleep due to pain, and presence of migraine symptoms (nausea/vomiting, phono/photophobia). Statistically significant differences between NVOP and OFM were found in the pain quality mimicking dental pain (χ^2^ = 24.2, df = 1, *p* < 0.001) and the presence of concomitant headache during attacks (NVOP 0%, OFM 61%, χ^2^ = 16, df = 1, *p* < 0.001). Cranial autonomic symptoms (CAS) were more frequently found in OFM (35%) than in NVOP (0%, χ^2^ = 7, df = 1, *p* = 0.008), (Table 1).

### Treatment Response

The sample size with adequate data for abortive therapy was small (n = 15, triptans = 11, non-steroidal anti-inflammatory drugs = 4) and analyses were therefore not performed. Data for prophylactic therapy was of adequate size for analysis of group effect (n = 25) but the individual groups were too small to allow inter-drug comparisons (amitriptyline = 16, topiramate = 4, valproate = 4, propranolol = 1). Based on pain diaries the sample was divided into two subgroups according to prophylactic response: ≥50% (good) and <50% (none-mild) reflecting improvement in frequency or severity of pain. These groups were used for further analysis.

There were no significant differences in response (see Table 2) to prophylactic pharmacotherapy between NVOP and OFM. Both groups demonstrated a very favorable response with up to 80% reporting significant improvement (see Table 2). All the remaining variables studied; gender, age, a throbbing quality to pain, waking from sleep due to pain, headache presence, migraine symptoms, CAS, pain quality mimicking dental pain and VAS at presentation, had no significant effect on treatment outcomes.

## 4. Discussion

Migraine is a complex neurovascular disorder characterized by recurrent episodes of moderate to severe headache, often accompanied by sensory disturbances and autonomic dysfunction. While traditionally associated with cranial pain, migraine-related mechanisms also play a significant role in orofacial pain syndromes, including orofacial migraine (OFM) and neurovascular orofacial pain (NVOP). OFM and NVOP are migraine-related entities affecting the facial and orofacial regions and have been individually classified in the International Classification of Orofacial Pain, 1st edition (ICOP) [1]. These two classifications, while distinct, exhibit significant overlap in their clinical characteristics. Numerous studies have highlighted the similarities in their presentation, leading to initial debates about whether they should be classified as separate entities. However, distinct diagnostic features do exist [7], supporting the ICOP classification that differentiates OFM from NVOP [1].

See Table 3 for summary. While these distinctions are established clinically, an essential question remains: Do OFM and NVOP differ in their response to pharmacotherapy?

At the core of the question as to whether OFM and NVOP would respond differently to pharmacotherapy lie specific pathophysiologic features. It is thought that migraine headache is a manifestation of a brain state of altered excitability capable of activating the trigeminovascular system in genetically susceptible individuals [16]. When peripheral anatomy remains insufficient to explain a low prevalence of facial pain presentation, such somatotopic segregation may be rather central. The central somatotopy of the trigeminal nucleus caudalis (sTN) is onion-ring-shaped with the center being the perioral region. However, fibers from the V1 branch project more to the caudal part of the trigeminal nucleus caudalis (sTN), whereas those from the V2 and V3 branches project more to the rostral part of the sTN [17]. This distribution also provides the anatomical basis of why cervically targeted therapies, e.g., greater occipital nerve (GON) block, may be effective in aborting headache disorders, since the V1 dermatome is projected to the most caudal part of the sTN and is located directly adjacent to the secondary sensory neuron of the C2/C3 branches in the spinal cord [18,19]. Similar evidence would be useful in explaining pain in the V2 and V3 dermatomes. Hypothetically, the facial presentations could be a simple “spread” of the pontomedullary activation in V2 and V3 facial presentations of headache, whereas the isolated facial attacks resembling headaches are due to a (extremely rare) direct functional connection between the limbic system and the maxillary or mandibular brainstem nuclei. Further studies into this subject are clearly needed. Functional imaging studies for headache and facial pain disorders suggested possible different mechanisms behind headache and facial pain. Brain activation in the sTN via trigeminal nociception was decreased in migraine [20] but increased in primary facial pain disorders (i.e., persistent idiopathic facial pain) [21] suggesting a role of hyperresponsive secondary sensory neurons in facial pain. Patients with an orofacial presentation of primary headache disorders have yet to be investigated using neuroimaging methods.

The results of pharmacotherapy in these entities have not been examined. This is the first study on the response to pharmacological treatment in NVOP and OFM. The existing literature, based largely on case reports, suggests that the response to pharmacological treatment is largely similar for OFM and NVOP [8,9,10,11,12,13,14,15]. A randomized, double-blind, placebo-controlled study investigating the efficacy and tolerability of sumatriptan in patients presenting with suspected migraine-related “sinus headache” demonstrated a significant response compared to placebo [12]. In this cohort, 69% of patients treated with a single 50 mg dose of sumatriptan achieved a positive headache response at 2 h, whereas the placebo response rate was 43% [12]. Furthermore, in a separate study, 81% of similar patients experienced at least a 50% reduction in pain with triptan use [11]. These findings provide compelling evidence that triptans are effective in treating migraine-related orofacial pain.

In patients with isolated migraine-related orofacial pain, triptan medications—including rizatriptan, zolmitriptan, and eletriptan—were prescribed for at least three consecutive attacks, with all patients reporting significant pain relief following triptan administration [9]. These findings reinforce the hypothesis that orofacial pains resembling primary headaches, such as OFM and NVOP, share common pathophysiological mechanisms with classic migraine. A systematic review examining the use of triptans for “sinus headache” concluded that a positive response to triptans in the absence of significant acute inflammatory findings strongly supports a diagnosis of migraine [10]. When these results are considered collectively, the utilization of abortive triptan therapy appears to serve as a diagnostic indicator for migraine-type orofacial pain. However, the therapeutic response does not distinguish between OFM and NVOP, despite the demarcation between these two conditions based on clinical characteristics. The question then arises as to the usefulness of further subclassification of migraine. There is consistent misdiagnosis of facial migraines. When pain is primarily in the midface area over the maxillary sinuses, a misdiagnosis of sinusitis is common. In contrast, pain in the lower part of the face is often related to dental pathology. These misdiagnoses could be minimized by a careful phenotyping of facial migraines. This would justify a separate classification for OFM.

While extensive research has been conducted on abortive treatment options, fewer studies have explored the effects of prophylactic pharmacotherapy on OFM and NVOP. Our study contributes to this gap in knowledge, demonstrating that both OFM and NVOP respond similarly to prophylactic antimigraine therapy. This suggests that, despite their distinct clinical presentations, these conditions share a common neurovascular pathophysiological basis. The efficacy of preventive treatments such as beta-blockers, calcium channel blockers, anticonvulsants (e.g., topiramate), and CGRP antagonists in both OFM and NVOP further supports this hypothesis. Additionally, the similar response of these conditions to serotonin receptor agonists and anti-inflammatory medications strengthens the argument that their underlying mechanisms are more alike than different.

Despite these findings, it remains uncertain whether the duration of prophylactic therapy should differ between OFM and NVOP patients. While some patients achieve long-term remission with continued prophylaxis, others experience recurrent attacks despite sustained treatment. Understanding the role of individual variability in treatment response is crucial, and future research should aim to identify predictive markers of therapeutic efficacy.

Eventually, the clinical utility of differentiating between OFM and NVOP from a pharmacotherapeutic perspective remains uncertain. While diagnostic clarity is crucial for classification purposes, the overlap in treatment response suggests that current pharmacologic strategies may be effective across both conditions. Future research should aim to refine our understanding of the molecular and neurovascular pathways involved in OFM and NVOP, potentially identifying biomarkers that could guide personalized treatment approaches.

This study has several limitations. The relatively small sample size may limit the generalizability of our findings, and larger, multicenter studies are needed to confirm our results. Additionally, the retrospective nature of the study introduces potential biases in data collection and patient recall. While our analysis focused on pharmacologic response, other factors such as lifestyle modifications and adjunctive therapies were not systematically evaluated. Lastly, long-term follow-up data were not available for all patients, making it difficult to assess sustained therapeutic efficacy over time.

While this study provides valuable insights into the pharmacotherapeutic response of OFM and NVOP, several areas warrant further investigation. Future research should prioritize larger, multicenter studies to improve the generalizability of findings and minimize biases associated with smaller sample sizes. Longitudinal studies with extended follow-up periods are essential to assess the long-term efficacy, safety, and sustainability of both abortive and prophylactic treatments. Additionally, well-designed randomized controlled trials (RCTs) are needed to establish evidence-based treatment protocols and determine the comparative effectiveness of different pharmacological interventions. Further exploration of genetic predispositions and neurovascular mechanisms may enhance our understanding of these conditions. Investigating the role of lifestyle factors, comorbid conditions, and non-pharmacological treatments—such as complementary medicine—could also contribute to a more comprehensive and personalized management strategy.

To conclude, our findings indicate that despite the distinct clinical characteristics of OFM and NVOP, both conditions exhibit a comparable response to pharmacotherapy. This reinforces the notion that they may share underlying pathophysiological mechanisms, warranting further investigation into targeted therapeutic strategies that could optimize patient outcomes. By advancing our understanding of pharmacologic interventions, clinicians can better tailor treatments to individual patients, improving both short-term pain relief and long-term disease management.

## 5. Study Limitations

Starting data were unavailable, precluding a more precise pharmacological analysis.

The small sample size limits any far-reaching conclusions. This is particularly true regarding individual drug efficacy. We were unable to analyze drug and dose responses separately due to data constraints.

## Figures and Tables

**Table 1 biomedicines-13-00714-t001:** Clinical characteristics of NVOP and OFM patients.

Parameter	NVOP	OFM	Univariate *p*-Value
Sex (M:F)	3:14	4:19	-
Baseline VAS (±SD)	7.7 ± 1.8	8.6 ± 1.3	-
Mean Age (yrs ± SD)			-
Male (yrs)	34.3 ± 17.6	35 ± 14.4	
Female (yrs)	41.3 ± 12.3	37.7 ± 18	
Laterality			0.11
Unilateral (%)	7 (41.2)	17 (73.9)	
Bilateral (%)	7 (41.2)	4 (17.4)	
Alternates (%)	3 (17.6)	2 (8.7)	
Pulsating (%)	7 (43.8)	11 (50)	-
Wakes (%)	5 (31.3)	10 (43.5)	0.33
Migraine Symptoms (%)	10 (62.5)	11 (47.8)	0.74
Headache			**<0.001**
Yes (%)	4 (23.5)	21 (91.3)	
None (%)	13 (76.5)	2 (8.7)	
Mimics Toothache (%)	17 (100)	5 (21.7)	**<0.001**
CAS (%)	0 (0)	8 (34.8)	**0.008**

Legend: n = number, NVOP = neurovascular orofacial pain, OFM = orofacial migraine, VAS = visual analog scale, migraine symptoms = nausea/vomiting and photo-/phono-phobia, headache = reports of associated migraine headaches, CAS = cranial autonomic symptoms, - = no statistics performed.

**Table 2 biomedicines-13-00714-t002:** Treatment outcome to prophylactic pharmacotherapy in NVOP and OFM patients.

Parameter	Treatment Outcome		*p*-Value
	None-Mild	Good	
**Diagnosis n (%)** NVOP OFM	3 (20) 9 (30)	12 (80) 16 (70)	0.48
**NVOP and OFM samples collapsed into one**			
**Sex n (%)**			-
Male	1 (10)	5 (17.9)	
Female	9 (90)	23 (82.1)	
**Baseline VAS (±SD)**	8.8 ± 1.5	8 ± 1.6	0.26
**Mean Age (yrs ± SD)**	40.6 ± 16.8	38.9 ± 15.4	0.99
**Laterality** n (%)			-
Unilateral	7 (70)	16 (57.1)	
Bilateral	3 (30)	7 (25)	
Alternates	0 (0)	5 (17.9)	
**Pulsating n (%)**	4 (40)	13 (50)	-
**Wakes n (%)**	5 (50)	9 (33.3)	-
**Migraine Symptoms n (%)**	4 (60)	7 (40)	-
**Headache** Yes (%) None (%)	3 (30) 7 (70)	18 (64.3) 10 (35.7)	0.74
**Mimics Toothache (%)**	3 (50)	17 (53.1)	-
**CAS (%)**	4 (40)	4 (14.8)	0.60

Legend: n = number, NVOP = neurovascular orofacial pain, OFM = orofacial migraine, VAS = visual analog scale, CAS = cranial autonomic symptoms, - = no statistics performed.

**Table 3 biomedicines-13-00714-t003:** Differences in diagnostic criteria and clinical features.

Feature	OFM	NVOP
**Primary Pain Location**	Jaws, Mid-face, Temple	Perioral, Intraoral
**Pain Character**	Pulsating,	Toothache-like, Pulsating,
**Severity**	Moderate to Severe	Moderate to Severe
**Triggers**	Physical activity, Stress	Cold food, Chewing
**Associated Symptoms**	Mild cranial autonomic symptoms, Nausea, Photophobia, Phonophobia	Mild cranial autonomic symptoms
**Headache Presence**	Common (66.7%)	Less common (30.8%)
**Responsiveness to treatment**	Similar	

## Data Availability

The original contributions presented in this study are included in the article. Further inquiries can be directed to the corresponding author.

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
