# Peer review of "Orofacial Migraine and Neurovascular Orofacial Pain: Response to Treatment—A Pilot Study"

_biomedicines, 2025, doi:10.3390/biomedicines13030714_

Round 1
Reviewer 1 Report
Comments and Suggestions for Authors
This work continues a line of research studied by this group who have published works on this topic already in the years 2023 and 2024. This latest work is aimed at evaluating the response to drugs of these two nosographic entities.
Line 43 and following: Is there an initial assumption that should be discussed and justified, also in light of what has already been discussed in references 11 and 12 to illustrate what the difference is with the definition of migraine without aura (ICHD 1.1)? And, also in light of the reported results, what would be the clinical usefulness of this further sub-classification? See also line 147 and following.
Line 198 and following. In the conclusions of this work there should be space for a reflection on the operational meaning of a further classification of headaches in addition to the ICHD. The data reported in this study, with the limitations recognized by the authors on the smallness of the sample, should lead to caution in subclassifying a pain syndrome that has characteristics that are not easily distinguishable from migraine, (1.1 vs OFM) and that respond equally to the same drugs, even specific ones such as triptans. This aspect needs to be discussed.
Table 1 lines 90: Explain what migraine symptoms consist of, and what the Headache line refers to.
Line 238 and following. The greatest methodological limitation seems to me to be linked to the fact that although the title of the work is the evaluation of response to drugs, data related to the type and dose of prophylactic drugs and an evaluation of triptans in relation to the two classes of patients examined (OFM and NOVP) are not reported. This aspect must be included in the reported analyses. Since the study is retrospective, I believe that a further analysis that divides the response to prophylactic treatments based on the main classes and their duration should be carried out and the data tabulated and included in the work to give greater value to the research.
Reviewer 2 Report
Comments and Suggestions for Authors
The authors should detail the specific pharmacotherapy, including the name of the medication, dose, and frequency. It is difficult to assess the response of NVOP and OFM to the pharmacotherapy of both drugs. Because abortive and prophylactic medication have different pharmacological mechanisms. If the authors want to investigate the accuracy of responses to pharmacotherapy, it is better to separate the two kinds of drugs and observe the response of NVOP and OFM to each drug.
Round 2
Reviewer 1 Report
Comments and Suggestions for Authors
The authors have responded sufficiently to the initial observations. As for the methodological limitation related to the lack of pharmacological evaluation, I believe that the response is insufficient. Not due to the fault or lack of will of the authors, but due to the impossibility of having the starting data. To ensure the possibility of publication, I therefore recommend adding an additional paragraph at the end of the discussion that underlines this limitation of the study. This limitation should also be reported in the abstract. I also recommend changing the title of the study to this: “Orofacial Migraine and Neurovascular Orofacial Pain: Response to treatment. A pilot study.”
Author Response
Reviewer 1
Comment: The authors have responded sufficiently to the initial observations. As for the methodological limitation related to the lack of pharmacological evaluation, I believe that the response is insufficient. Not due to the fault or lack of will of the authors, but due to the impossibility of having the starting data. To ensure the possibility of publication, I therefore recommend adding an additional paragraph at the end of the discussion that underlines this limitation of the study. This limitation should also be reported in the abstract. I also recommend changing the title of the study to this: “Orofacial Migraine and Neurovascular Orofacial Pain: Response to treatment. A pilot study.”
Response: We thank the reviewer.
As suggested we have added a limitations section after the discussion (lines 310-314)
“Study Limitations
Starting data was unavailable precluding a more precise pharmacological analysis.
The small sample size limits any far reaching conclusions. This is particularly true re-garding individual drug efficacy. We were unable to analyze drug and dose responses separately due to data constraints.”
Also added a comment in the abstract (lines 30-33).
“Study limitations include the lack of starting data precluding a more precise pharma-cological analysis. The small sample size limits any far reaching conclusions. This is particularly true regarding individual drug efficacy. We were unable to analyze drug and dose responses separately due to data constraints.”
The title has been changed as requested.
Reviewer 2 Report
Comments and Suggestions for Authors
I suggest the authors supply more sample and separate the two kinds of drugs and observe the response of NVOP and OFM to each drug. Too small sample and more drugs using can make the conculsion unresionable.
Author Response
Reviewer 2
Comment: I suggest the authors supply more sample and separate the two kinds of drugs and observe the response of NVOP and OFM to each drug. Too small sample and more drugs using can make the conculsion unresionable.
Response: The reviewer is of course correct in his statement that the sample size is too small for individual drug analyses. Nevertheless the results are interesting in the finding that ‘as a whole’ OFM and NVOP respond the equally to prohylactic therapy.
As detailed in response to reviewer 1 we have added a limitations section which addresses the comment of reviewer 2 above.